# Seismic Behavior of Steel Plate-Concrete Shear Walls with Holes

**Zhihua Chen [1,2], Jingshu Wu [2,3,\*] and Hongbo Liu [1,2]**

[1] Department of Civil Engineering, Tianjin University, Tianjin 300072, China; zhchen@tju.edu.cn (Z.C.); hbliu@tju.edu.cn (H.L.)
[2] State Key Laboratory of Hydraulic Engineering Simulation and Safety, Tianjin 300072, China
[3] Research Institute of Building and Construction CO.LTD.MCC Group, Beijing 100088, China
\* Correspondence: wujingshu@nqstc.com

**Abstract:** Steel plate-concrete shear walls (SPSW) are used as the containment in nuclear power stations. However, the influence of holes and axial loading on the behavior of steel plate-concrete shear walls is neglected in most studies. Thus, it is necessary to understand the seismic behavior of SPSW members with holes in order to avoid the potential risks for nuclear power stations. In this study, a series of specimens were tested by low-cycle reciprocal loading. The specimens were designed with different holes to simulate real members in nuclear power stations. A hysteretic curve of specimens was obtained from a low-cycle reciprocal test to discuss the seismic behavior of steel plate-concrete shear walls (SPSW). Moreover, effects of axial compression ratio, hole size, thickness of the steel plate, and hole position on the hysteretic performance of SPSW were analyzed. The horizontal ultimate bearing capacity of SPSW specimens was estimated using the norms of the Architecture Institute of Japan and the calculation method of Ono reduction rate. Results provide theoretical references for the design and application of SPSW with holes.

**Keywords:** SPSW; axial compression ratio; hole size; thickness of stiffening plate; hole position; hysteretic performance

## 1. Introduction

Steel plate-concrete shear walls (SPSW) serve as the containment of AP1000 and CAP1400 nuclear power stations and of the stress components of the internal plants in nuclear power stations. Structures with SPSW have advantages of modular construction and strong seismic and impact resistance. In the 1960s, Japan began to apply the steel stiffening concrete seismic structural structure. A series of studies on shear capacity and the stiffness and ductility of SPSW structures have been conducted in many countries, such as the US and Japan [1]. Other relevant research is based on low-cycle reciprocal tests. Lubell et al. [2] conducted an experimental testing on two single and one four-story steel shear wall specimens under cyclic quasi-static loading. Hjjar et al. [3] performed a low-cycle reciprocal test on a 1:2 scale model of SPSW structure and found its high bearing capacity, energy-consumption mechanism, and good ductility. The test results show that the superposition principle is basically true [4]. Berman et al. [5] conducted an experiment on the light-gauge steel plate shear walls and braced frames to study the hysteretic behavior. They revealed that the energy dissipated per cycle and the cumulative energy dissipation are similar for the two structures. Later, Gan, Zhao, and Wang et al. [6–8] studied the seismic behavior of the steel plate-reinforced concrete shear wall by using the quasi-static test. Li and li [9] investigated the out-of-plane seismic behavior of steel plate and concrete infill composite shear walls (SCW). They found that SCW has a better ultimate capacity and lateral stiffness. Huang et al. [10] proposed an innovative concrete-filled double-skin steel plate SCW

and investigated its seismic behavior. By conducting a quasi-static cyclic test, the wall is confirmed to have good seismic performance.

In recent years, various scholars have simulated the hysteretic curve and stiffness reduction rate of SPSW through finite element simulation. Rafiei et al. [11] presented and verified the finite element model to simulate the behavior of a novel SCW consisting of the two skins of profiled steel sheeting and an infill of concrete under in-plane loadings. Hu et al. [12] analyzed the moment-curvature behavior of concrete-filled steel plate SCW using refined material constitutive models. Peter et al. [13] presented the development and benchmarking of a detailed 3D nonlinear inelastic finite element model to predict the lateral load-deformation response and behavior of the 1/6th scale test structure. Nguyen et al. [14] presented a numerical study of steel-plate concrete composite walls by using the general-purpose finite element program ABAQUS. The influence of key design variables, including the reinforcement ratio, connector type, and faceplate slenderness ratio, were likewise studied. Wang et al. [15] investigated the hysteretic performance of the SPSW wall by using Open Sees software. Moreover, parameters such as the steel plate ratio, axial compressive load ratio, concrete strength, and web reinforcement ratio were analyzed. Yamatani [16] performed a low-cycle reciprocal test of SPSW with holes at the lower position under the shear-span ratio of 0.7, a distance–thickness ratio of 100, and opening area ratio of 0.3. Other scholars proposed the concept of reduction coefficient to evaluate the bearing capacity of an SPSW wall. Satou Kouichiet al. [17] conducted a numerical simulation of an SPSW structure and found that a numerical simulation is applicable for the calculation and analysis of the performance of shear walls with holes. Ishida Masatoshi [18] analyzed the seismic behavior of an SPSW structure by using theories and finite element simulation. Fujita Tomohiro and Oosuka et al. [19,20] performed anti-shear tests on SPSW structures with holes. Adding sleeves and using increased thickness surface steel plates on the shear wall were determined to be effective reinforcement methods. Some scholars also used the XFEM (the eXtended Finite Element), XIGA (the eXtended IsoGeometric Analysis) and Jaya algorithm to predict the occurrence of cracks and other defects in walls and slab [21–23].

So far, steel plate-concrete shear walls are studied widely. However, the influence of holes and axial loading on the behavior of steel plate-concrete shear walls are neglected in most studies. Thus, the seismic behavior of steel plate-concrete shear walls is completely different when the influences are considered. In the study, a series of low-cycle reciprocal loading tests are conducted on an SPSW structure with holes, thus obtaining the ultimate bearing capacity and failure mode of the structure. The influences of holes and reinforcing measures and axial loads of components on the seismic behavior of an SPSW structure are analyzed, thus determining the difference between theoretical and test values. The seismic behavior and stress mechanism of SPSW specimens are discussed theoretically.

## 2. Experiment

### 2.1. Experimental Apparatus

A low-cycle reciprocal test of an SPSW structure with different hole sizes was conducted in the Beijing Key Laboratory of Engineering Anti-earthquake and Structural Diagnosis of the Beijing University of Technology. The test applied a horizontal servo actuator (maximum range = 2000 kN), which was fixed on the concrete counterforce wall. Test devices included a counterforce wall, counterforce frame, loading beam, bottom beam, and displacement meter. The bottom beam was strongly connected to the ground through a fixing device, and a steel ingot was used on the loading beam. On the one hand, this design prevents concrete crushing caused by excessive force. On the other hand, the vertical load is ensured to be a uniform load. In the test, the low-cycle reciprocal loading test device is composed of horizontal and vertical jacks. In this device, the jack on the transverse counterforce frame applies the vertical loads. During the whole test, these vertical loads are kept as stable as possible through the manual control of the oil pump. The horizontal jack applies the horizontal loads. One end of the jack is connected to the loading beam of the specimens, and the other end is fixed on the counterforce wall (Figure 1).

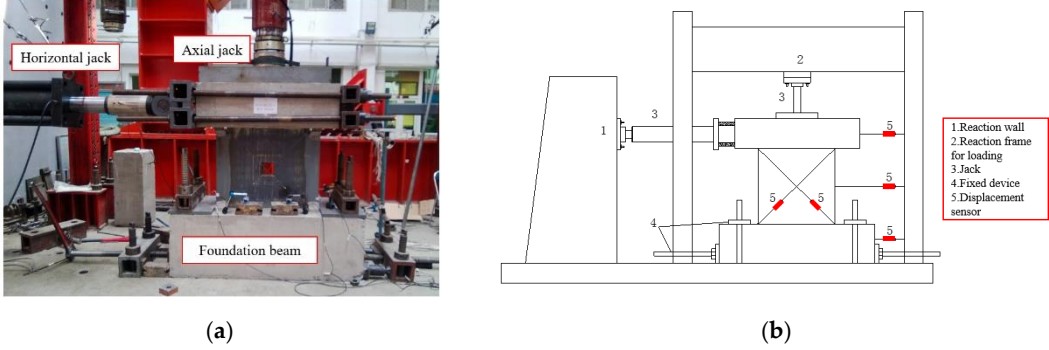

**Figure 1.** Loading device (**a**) and measurement arrangement (**b**).

*2.2. Design of Specimens and Instrumentation*

To accurately reflect shielding plants and internal SPSW structures in a nuclear power station, specimens in this study were designed with reference to Japan's *Technical Regulations on Earthquake-resistant Design of Steel Plate Concrete Structure* (JEAC4618-2009) [24] and design documents of AP1000 demonstration projects. According to *Technical Regulations on Earthquake-resistant Design of Steel Plate Concrete Structure*, the ratio between the thickness of SPSW for nuclear safety and thickness of the surface steel plate should be controlled within 30:200. On this basis, thicknesses of the shear wall and surface steel plate were set. Considering the actual welding ability in the laboratory, the thickness of the surface steel plate in the test specimens was determined to be 2.5 mm.

In this study, low-cycle reciprocal tests of nine SPSW specimens were conducted. Dimensions of SPSW specimens were 800 mm (height) × 800 mm (width) × 125 mm (thickness). SPSW specimens were divided into three types: (1) without holes, (2) with small holes, and (3) with large holes. Considering the loading capacity of the laboratory, the SPSW specimens without holes and with small holes were flat (Figure 2), whereas the SPSW specimens with large holes were I-shaped (Figure 3). The stiffening plates were spread around the hole, with a width is 90 mm, and the length determined by the perimeter of the hole. The hole size, thickness of the stiffening plate, hole position, and axial pressure ratio are listed in Table 1. In the SPSW specimens, the steel plate was made of Q235B, and the grade of the concrete strength was C35. Material properties are shown in Tables 2 and 3. Fine aggregates with a diameter smaller than 10 mm were used.

**Table 1.** Parameters of specimens.

| Specimen | Flange Size (mm) | | Thickness of Steel Plate (mm) | Hole Size (mm) | Hole Position | Thickness of Stiffening Plate | Axial Compression Ratio | Wall Type |
|---|---|---|---|---|---|---|---|---|
| | Width | Thickness | | | | | | |
| SCW-1 | – | – | 2.5 | – | – | – | 0.3 | Flat |
| SCW-2 | – | – | 2.5 | 125 × 125 | Center | – | 0.15 | Flat |
| SCW-3 | – | – | 2.5 | 125 × 125 | Center | 2.5 | 0.15 | Flat |
| SCW-4 | – | – | 2.5 | 125 × 125 | Center | 2.5 | 0.3 | Flat |
| SCW-5 | – | – | 2.5 | 125 × 125 | Center | 2.5 | 0.5 | Flat |
| SCW-6 | – | – | 2.5 | 125 × 125 | Center | 3.5 | 0.3 | Flat |
| SCW-7 | – | – | 2.5 | 180 × 180 | Center | 2.5 | 0.15 | Flat |
| SCW-8 | 395 | 110 | 2.5 | 430 × 600 | Center | 2.5 | 0.15 | I-shaped |
| SCW-9 | 395 | 110 | 2.5 | 430 × 600 | Eccentricity | 2.5 | 0.15 | I-shaped |

**Table 2.** Mechanical properties of the steel plate.

| Thickness of Steel Plate (mm) | Yield Strength (MPa) | Tensile Strength (MPa) | Elasticity Modulus (MPa) | Maximum Elongation (%) |
|---|---|---|---|---|
| 2.5 | 369 | 486 | $2.02 \times 10^5$ | 28.5 |

**Table 3.** Mechanical properties of the concrete.

| Material Number | Axial Compressive Strength (MPa) | Axial Tension Strength (MPa) | Elasticity Modulus (MPa) |
|---|---|---|---|
| C35 | 25.98 | 2.33 | $3.25 \times 10^4$ |

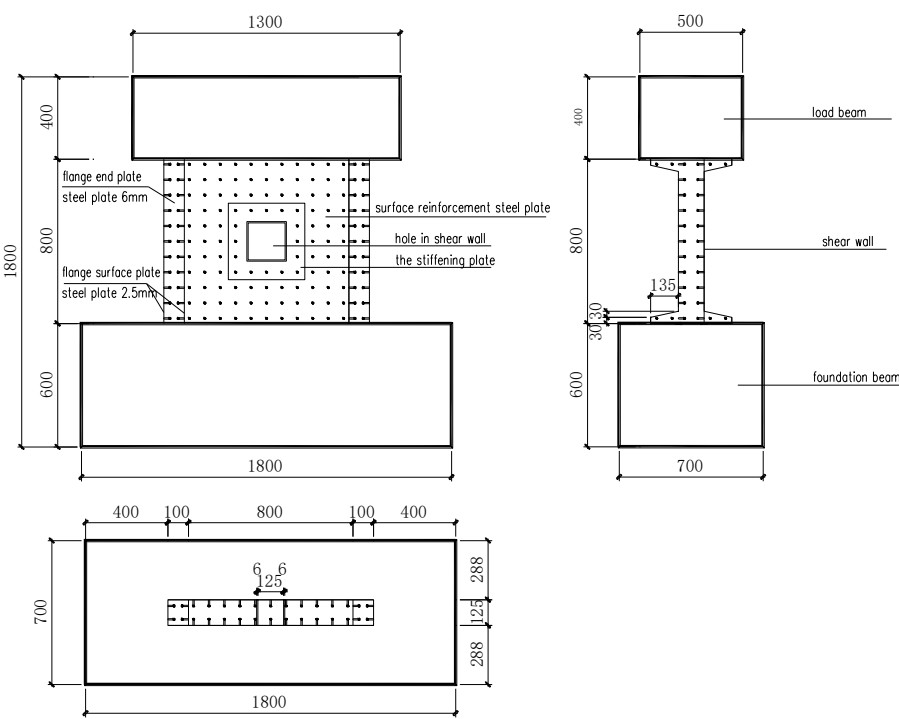

**Figure 2.** Typical flat specimens.

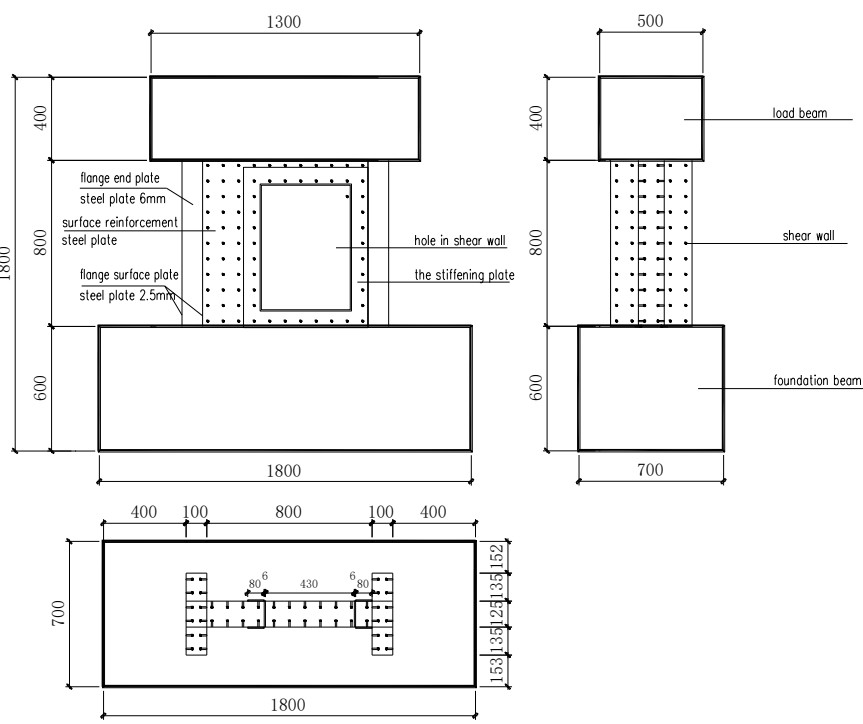

**Figure 3.** Typical I-shaped specimens.

In the test, five displacement sensors were set (Figure 1), three of which were set vertically on the wall. The first displacement sensor on the side of the loading beam of the shear wall tested the top wall displacement under reciprocal loads. The displacement sensor in the center of the shear wall examined the bottom slippage during the loading process. The displacement sensor on the side of the bottom beam tested the overall slippage of specimens to assure test accuracy. In addition, two dial indicators were installed at diagonal positions of the wall to test for outward deflection.

### 2.3. Loading Mode

The loading process is composed of pre-loading and formal loading. First, the vertical jack applied the vertical loads, and then 20kN force was pre-loaded by a horizontal actuator. Second, loads were removed to ensure contact of the loading device with the specimens. The loading protocol refer to construction standards of the China Construction Industry JGJ/T 101-2015 [25].

In the formal loading, the vertical jack also applied the vertical loads on the specimens. Before full loading, the vertical loads were applied 2–3 times according to the design value of 40–60% to eliminate the influences of the internal non-uniformity of shear wall specimens. When the vertical load was applied to the designated value and stabilized, a horizontal actuator was used to apply the horizontal low-cycle reciprocal loads to the specimens. Load control was applied in the early stage of the test. Loads in the first, second, third, and fourth cycles were 50, 100, 200, and 400 kN, respectively. Later, loads were increased by 100 kN every cycle until specimens fail. The loading system is illustrated in Figure 4. Specimens were considered a failure upon attaining one of the following cases:

(1) The bearing capacity of specimens decreased to lower than 85%.
(2) Serious failure occurred in specimens, such as heaving of steel plates or crushing of concrete, thereby resulting in a difficult loading.

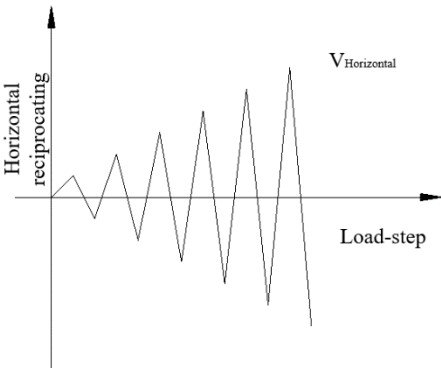

**Figure 4.** Loading system.

## 3. Experimental Analysis

### 3.1. Failure Mode

Specimen SCW-1 is an SPSW without holes. The loading process of SCW-1 was relatively stable. When the cyclic loading approaches 700 kN, the concrete begins to crush, and the steel plate develops slight deformation. With the continuous increase in loads, the crushing region in the concrete and deformation of steel materials likewise continued. Heaves develop on the steel plate at the root side of the shear wall until the load reaches 1100 kN. When the cyclic loads increase to 1240 kN, heaves at the root side of the steel plate of the shear wall and surface steel plate intensify, accompanied with the crushing of concrete at root positions. Due to the processing deviation of specimens and influences of initial defects, deformations of steel plates at the two root sides of the shear wall are inconsistent but are symmetric to an extent.

In the tests of flat specimens with small holes, the horizontal loads are small in the initial loading stage, and specimens are still elastic. A linear relationship is observed between the horizontal load and vertex displacement of specimens. In the late stage, internal concrete begins to crack as horizontal loads increase. The side and surface plates yield successively. As lateral displacement increases, the steel plate begins to bend or crack. The weld joints of several specimens are pulled open, and internal concretes are crushed, resulting in failure (Figure 5). The main failure develops at the wall bottom and is evaluated as brittle failure. Several specimens develop slippage due to the lower stiffness of the bottom beams.

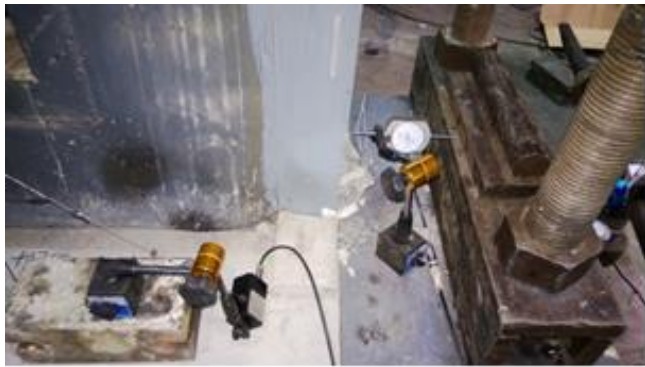 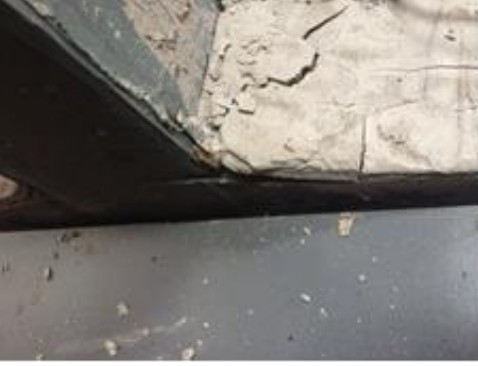

(**a**) buckling of side plates and pulling failure of weld joints          (**b**) concrete crushing

**Figure 5.** Bucklingof side plates, pulling failure of weld joints, and concrete crushing of flat steel plate-concrete shear wall (SPSW) specimens with small holes.

Figure 6 shows the failure mode of I-shaped SPSW specimens with large holes. Bending shear failure is the dominant failure mode. In the test of SCW-8, the internal concrete in SPSW begins to crack at approximately 500 kN. Furthermore, the steel plate at the holes begins to yield and pull open at approximately 800 kN. The surface steel plate yields at 1100 kN. Subsequently, the surface plate cracks and bends, accompanied with wrinkles. The internal concrete of the steel plate is crushed after losing constraints. Moreover, the steel plate at the wing wall side buckles, and the wall develops serious deformation. The bearing capacity of SPSW specimens continue to decrease, and the loading test is terminated. According to the results, SCW-8 with holes at the center and SCW-9 with holes biased to the axis have no significant differences in terms of bearing capacity and failure mode.

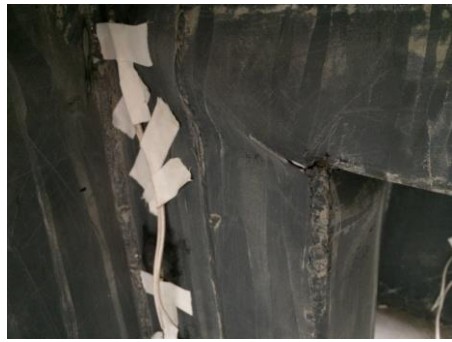 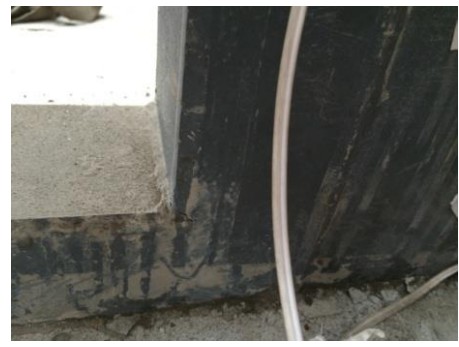

(**a**) top left of SCW-8                          (**b**) bottom right of SCW-8

**Figure 6.** *Cont.*

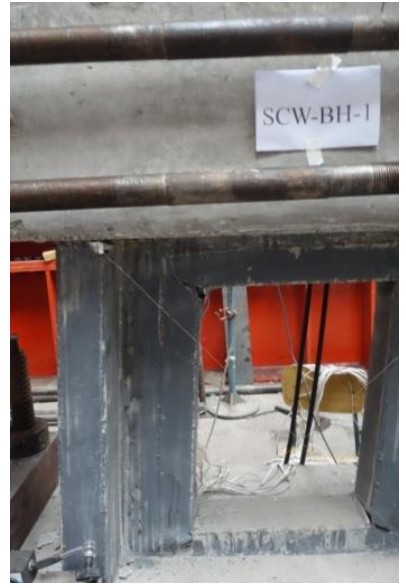
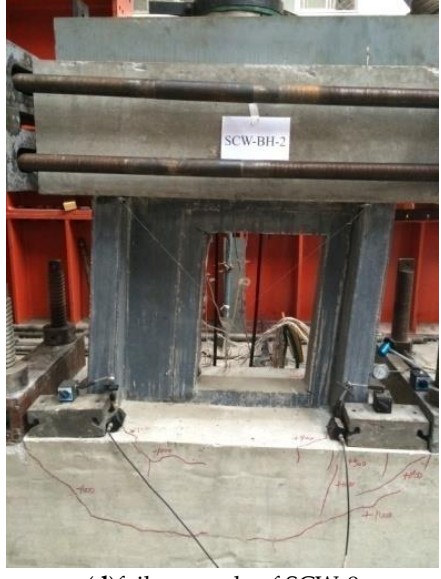

(**c**)failure mode of SCW-8          (**d**)failure mode of SCW-9

**Figure 6.** Failuremodes of I-shaped SPSW specimens with large holes.

### 3.2. Effects of Axial Compression Ratio on the Hysteretic Performance of SPSW Specimens

The axial compression ratio is an important factor that influences the hysteretic performance of SPSW specimens. In this section, axial compression ratios in the low-cycle reciprocal tests of three SPSW specimens with holes (SCW-3, SCW-4, and SCW-5) are set at 0.15, 0.3, and 0.5, respectively. The ultimate loads and displacements of specimens are shown in Table 4. Figure 7 illustrates the hysteretic curves of different SPSW specimens, whereas Figure 8 shows the skeleton curves.

**Table 4.** Ultimate loads and displacements of specimens.

| Specimens | Ultimate Loads (kN) | | Ultimate Displacements (mm) | |
|---|---|---|---|---|
| | Positive | Negative | Positive | Negative |
| SCW-3 | 1100 | −1074 | 18.09 | −27.05 |
| SCW-4 | 1217 | −1185 | 18.46 | −28.60 |
| SCW-5 | 1362 | −1301 | 16.56 | −13.88 |

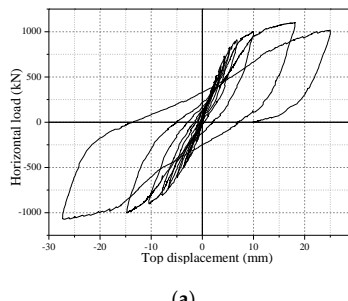
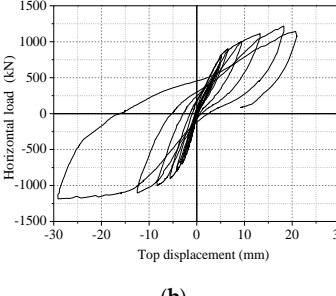
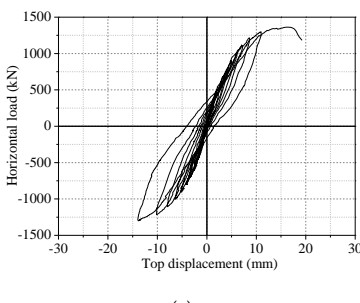

(**a**)            (**b**)            (**c**)

**Figure 7.** Hysteretic curve of specimens under different stress ratios. (**a**) SCW-3, (**b**) SCW-4, and (**c**) SCW-5.

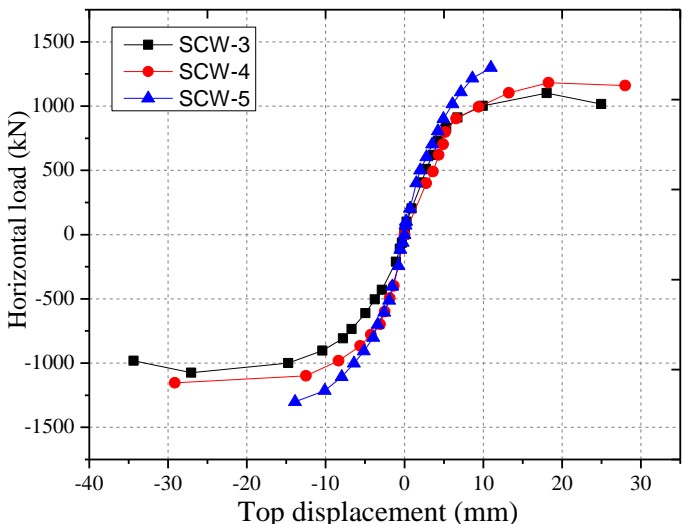

**Figure 8.** Comparison of the skeleton curves of specimens of the same size under different axial compression ratios.

In the loading process of the three specimens, steel plates develop low sounds, and internal concretes begin to crack at 350 kN when the axial compression ratio is 0.5. When the axial compression ratios are 0.15 and 0.3, steel plates develop low sounds at 500 kN and 150 kN, respectively. In summary, the concrete begins to crack early under large axial compression ratios. With the continuous increase of loads, SCW-3 and SCW-4 show a reduction stage of loads and good ductility. When loads increased to 1217 kN, the root plate of SCW-4 bent, and the test was terminated. SCW-4 thus shows poor ductility, as verified from the skeleton curve. Table 4 shows that the ultimate bearing capacity of components is positively related with axial compression ratio, but deformation resistance and ductility are negatively correlated. Therefore, attention should be paid to the sudden failure of structures under high axial compression ratio.

### 3.3. Effects of Hole Area on the Hysteretic Performance of SPSW Specimens

Influences of hole size on lateral bearing capacity, deformation resistance, and energy consumption of SPSW specimens with different hole sizes are determined through a low-cycle reciprocal test. Ultimate strengths and displacements of SPSW specimens with different hole sizes are presented in Table 5. Figures 9 and 10 respectively display the hysteretic and skeleton curves of SPSW specimens with different hole sizes.

**Table 5.** Ultimate loads and displacements of specimens.

| Specimens | Hole Size (mm) | Ultimate Loads (kN) | | Ultimate Displacements (mm) | |
|---|---|---|---|---|---|
| | | Positive | Negative | Positive | Negative |
| SCW-1 | – | 1277 | −1310 | 23.15 | −19.27 |
| SCW-3 | 125 × 125 | 1100 | −1074 | 18.09 | −27.05 |
| SCW-7 | 180 × 180 | 1190 | −1105 | 37.43 | −27.45 |
| SCW-8 | 430 × 600 | 1055 | −1007 | 51.94 | −43.49 |

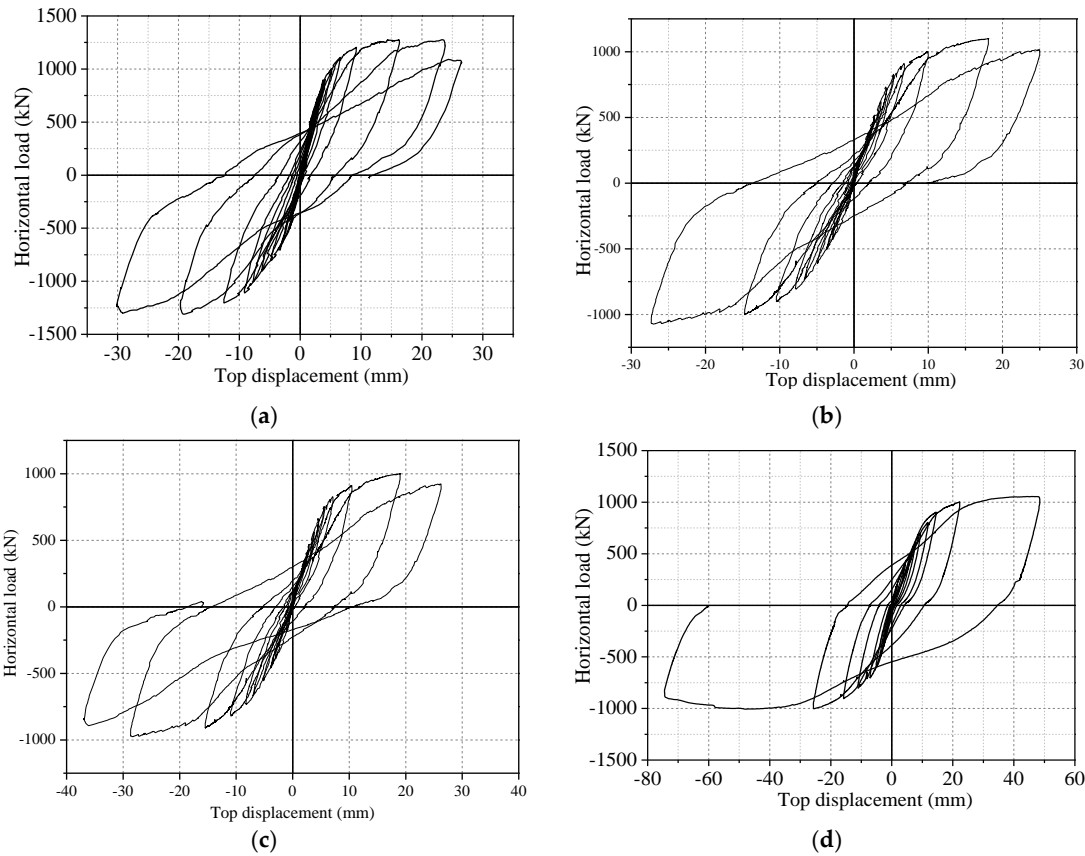

**Figure 9.** displacement curves of SPSW specimens with different hole sizes. (**a**) SCW-1, (**b**) SCW-3, (**c**) SCW-7, and (**d**) SCW-8.

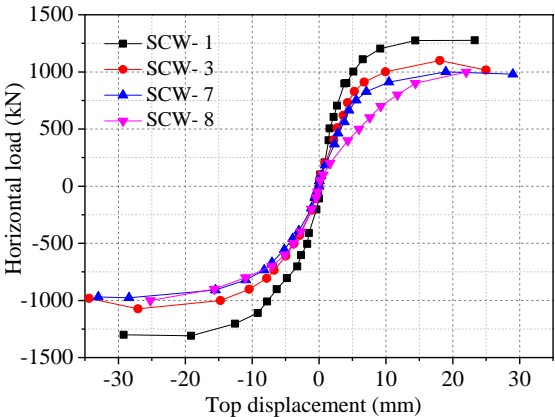

**Figure 10.** Skeleton curves of SPSW specimens with different hole sizes.

Table 5 shows that as hole area increases, the ultimate displacement of specimens gradually increases, but the ultimate loads of specimens decrease slightly. Hysteretic curves of SCW-1, SCW-3, and SCW-7 have similar shapes. They are full but have clear twist contraction effects. Compared with the first three specimens, SCW-8 has a fuller hysteretic curve, indicating better seismic behavior.

*3.4. Effects of the Thickness of the Stiffening Plate on the Seismic Behavior of SPSW Specimens*

SCW-1, SCW-4, and SCW-6 have the same appearance size.SCW-1has no hole, SCW-4 and SCW-6 have the same geometric size, hole position, and hole size but are equipped with 2.5 mm and 3.5 mm stiffening plates, respectively. Table 6 shows that compared with the uncut SCW-1, the strength and stiffness of the open specimens are decreased. Compared with the ultimate strength of SCW-4, the

ultimate strength of SCW-6 increased by 4% in the positive direction and 10% in the negative direction. The hysteretic curves of SCW-4 and SCW-6 are shown in Figure 11, and the skeleton curves of specimens are shown in Figure 12. The hysteretic curve of SCW-6 is S-shaped, which is attributed to slippage caused by anchoring failure. Similarly, the SCW-6 skeleton curve displays that its initial stiffness is low due to the large specimen displacement.

**Table 6.** Ultimate loads and displacements of specimens.

| Specimens | Ultimate Loads (kN) | | Ultimate Displacements (mm) | |
|---|---|---|---|---|
| | Positive | Negative | Positive | Negative |
| SCW-1 | 1277 | −1310 | 23.15 | −19.27 |
| SCW-4 | 1217 | −1185 | 18.46 | −28.60 |
| SCW-6 | 1260 | −1302 | 21.41 | −28.11 |

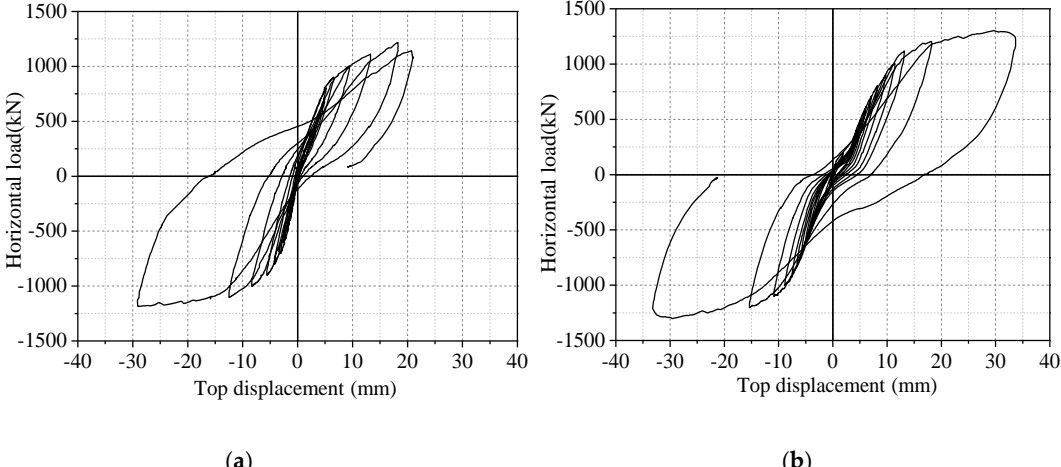

(**a**)  (**b**)

**Figure 11.** Load displacement curve of specimens reinforced by different thicknesses of plates. (**a**) SCW-4 and (**b**) SCW-6.

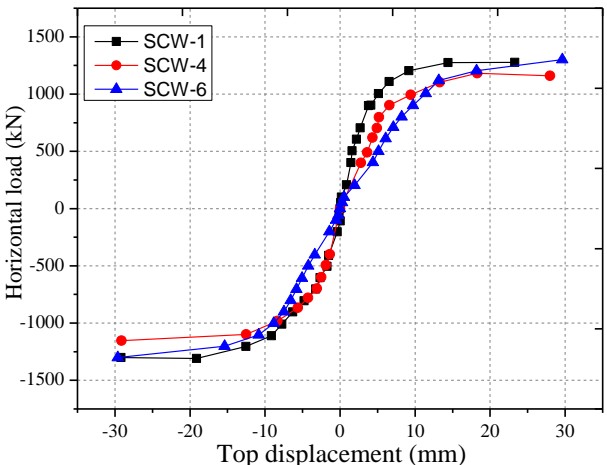

**Figure 12.** Skeleton curves of specimens reinforced by different thicknesses of plates.

*3.5. Effects of Hole Position on the Seismic Behavior of SPSW Specimens*

Figures 13 and 14 illustrate the hysteretic and skeleton curves of specimens with large holes at different positions as determined through low-cycle reciprocal tests, respectively. The ultimate strengths and displacements of specimens are presented in Table 7.

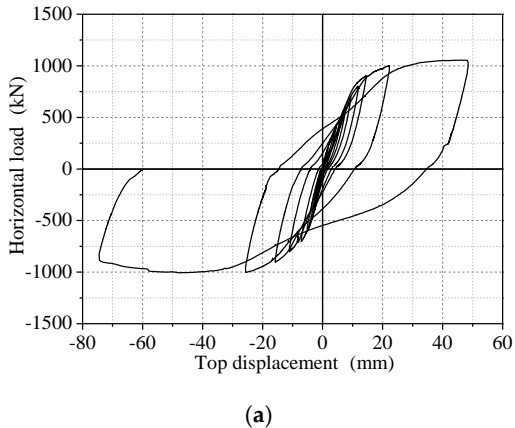
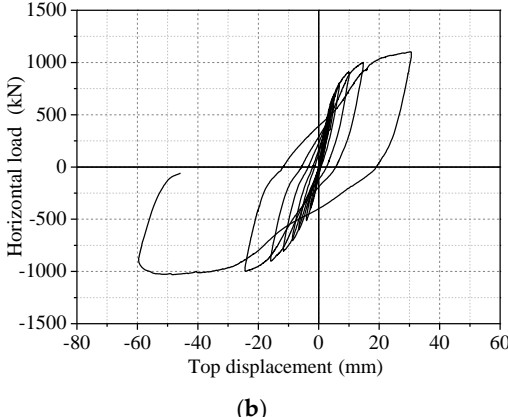

(**a**)　　　　　　　　　　　　　　　　　　　　　　　　(**b**)

**Figure 13.** Hysteretic curves of specimens with holes at different positions. (**a**) SCW-8 and (**b**) SCW-9.

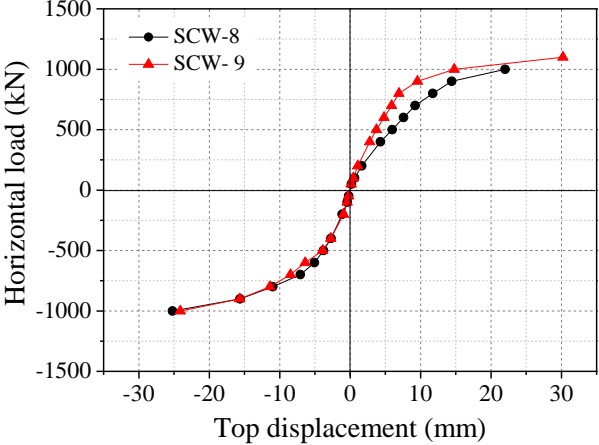

**Figure 14.** Skeleton curves of specimens with holes at different positions.

**Table 7.** Ultimate loads and displacements of specimens.

| Specimens | Ultimate Loads (kN) | | Ultimate Displacements (mm) | |
| --- | --- | --- | --- | --- |
| | Positive | Negative | Positive | Negative |
| SCW-8 | 1055 | −1007 | 51.94 | −43.49 |
| SCW-9 | 1103 | −1031 | 34.14 | −44.41 |

Table 7 shows that the ultimate displacement of the specimen with a hole in the center is higher than that of the specimen with an eccentric hole. Comparing the two specimens, the hysteretic curve of the specimen with a hole in the center is fuller, and its energy-dissipation capacity is larger. Hence, placing the hole in the center of SPSW structures is suggested for practical engineering.

## 4. Theoretical Calculation

In order to estimate the ultimate bearing capacity and lateral stiffness resistance of SPSW specimens with holes, two calculation methods are discussed. The first method is the reduction rate calculation method of the Architecture Institute of Japan (AIJ) [26], and the second method is the Ono method [27]. A comparison between the experimental results and the theoretical results are discussed.

### 4.1. ReductionRate Calculation Method of AIJ

The calculation methods of the reduction rates of horizontal bearing capacity ($r_u$) and stiffness ($r_c$) of shear wall components with holes are regulated in the structural design codes of AIJ. These two calculation formulas are as follows:

$$r_u = 1 - \eta, \tag{1}$$

$$r_c = 1 - 1.25 \sqrt{\frac{h_0 l_0}{hl}}, \tag{2}$$

where $h_0$ and $l_0$ are the height and width of the hole, $h$ is the floor height, $l$ is the center distance of frame columns, $\eta$ is the hole area ratio with value $\eta = \max \left\{ \sqrt{\frac{h_0 l_0}{hl}} \right\}$, and where $\sqrt{\frac{h_0 l_0}{hl}} \leq 0.4$. In AIJ codes, the application of these formulas is regulated; they are applicable to situations when the hole area is smaller than 0.4 but not for situations when the hole area is larger than 0.4. In these formulas, the shape and position of holes are not considered when calculating bearing capacity and stiffness. That is, the calculated results of bearing capacity and stiffness are the same for specimens with the same hole size.

### 4.2. Ono Reduction Rate Calculation

Ono Masasyuki, a Japanese scholar, proposed the calculation method of Ono reduction rate through a series of experiments wherein the influences of hole positions are considered [27] (Figure 15).

$$r_u = \sqrt{\sum A_{ei}/hl}, \tag{3}$$

when $0.1 \leq \gamma \leq 0.53$,

$$r_e = \left(\frac{0.025}{0.0303^\gamma}\right) \times \alpha\beta + \left(\frac{0.6}{0.55^\gamma}\right) / \left(\frac{1.55}{\gamma^2}\right)^k, \tag{4}$$

when $0.53 \leq \gamma \leq 1.00$,

$$r_e = \left(\frac{5.24}{717.24^\gamma}\right) \times \alpha\beta + \left(\frac{0.6}{0.55^\gamma}\right) / \left(\frac{1.55}{\gamma^2}\right)^k, \tag{5}$$

where $A_{ei}$ is the area of shadow part, $h$ is the floor height, and $l$ is the center distance of frame columns. $\alpha\beta = 2bD/tl'$; where $bD$ is the sectional area of columns, $t$ is the wall thickness, and $l'$ is the net span. $\gamma = 2l_w/l'$; and where $l_w$ is the length of the wall after the hole is removed. $\kappa = h_0/h'$, where $h_0$ is the height of the hole and $h'$ is the net height of the wall.

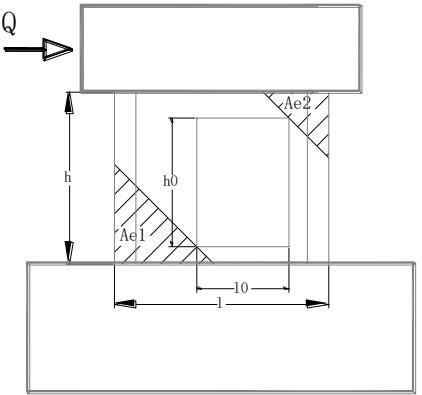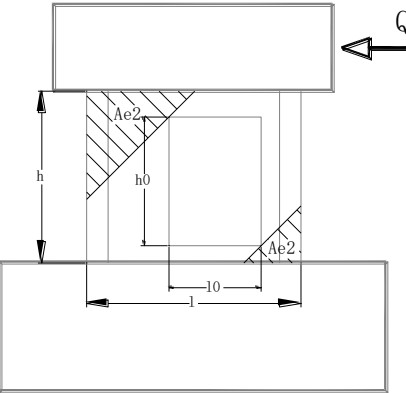

**Figure 15.** Effective supporting area in the Ono reduction rate calculation method.

In this study, the horizontal ultimate bearing capacity of SPSW members with holes can be obtained by multiplying the reduction rate by that of the corresponding members without holes. The

lateral stiffness of SPSW members with holes can be obtained with the same method. Change of the concrete compression zone is considered in the method of Ono reduction rates. The reduction rates are different in different directions for the specimen with biased holes.

Table 8 displays the comparison between the test results of horizontal ultimate loads and the calculated results based on strength reduction rate. Table 9 compares the test results of lateral stiffness and calculated results based on the stiffness reduction rate. This test has no effective lateral stiffness of I-shaped SPSW structures without holes, and only the comparisons between the calculated and test results of SCW-4 and SCW-6 are presented. According to the average ratio, the calculated value of the Ono formula is closer to the experimental value than that of the code formula.

**Table 8.** Comparison of ultimate loads.

| Specimen | Stress State | $Q_{test}$ /kN | $Q_{code}$ /kN | $Q_{code}/Q_{test}$ | $Q_{ono}$ /kN | $Q_{ono}/Q_{test}$ | Axial Compression/kN | Type of Walls |
|---|---|---|---|---|---|---|---|---|
| SCW-4 | Stressed | 1185 | 1127 | 0.95 | 1135 | 0.96 | 950 | Flat |
| | Tensioned | 1217 | 1098 | 0.90 | 1106 | 0.91 | | Flat |
| SCW-6 | Stressed | 1302 | 1127 | 0.87 | 1269 | 0.98 | 950 | Flat |
| | Tensioned | 1260 | 1098 | 0.87 | 1237 | 0.98 | | Flat |
| SCW-8 | Stressed | 1007 | 734 | 0.73 | 820 | 0.81 | 320 | I-shaped |
| | Tensioned | 1055 | 734 | 0.70 | 857 | 0.81 | | I-shaped |
| SCW-9 | Stressed | 1031 | 734 | 0.71 | 836 | 0.81 | 320 | I-shaped |
| | Tensioned | 1103 | 734 | 0.67 | 836 | 0.76 | | I-shaped |
| Average ratio | | | | 0.80 | | 0.88 | | |

Notes: $Q_{test}$ is the test result of horizontal ultimate loads. $Q_{code}$ and $Q_{ono}$ are horizontal ultimate bearing capacities calculated by AIJ codes and the Ono calculation formula.

**Table 9.** Comparison of lateral stiffness.

| Specimens | | $K_{test}$ | $K_{code}$ | $K_{code}/K_{test}$ | $K_{ono}$ | $K_{ono}/K_{test}$ | Axial Compression/kN |
|---|---|---|---|---|---|---|---|
| SCW-4 | Negative | 41.43 | 56.11 | 1.35 | 67.64 | 1.63 | 950 |
| | Positive | 65.90 | 45.51 | 0.69 | 54.87 | 0.83 | |
| SCW-6 | Negative | 46.31 | 56.11 | 1.21 | 67.64 | 1.46 | 950 |
| | Positive | 58.86 | 45.51 | 0.77 | 54.87 | 0.93 | |

Notes: $K_{test}$ is the test result of lateral stiffness. $K_{code}$ and $K_{ono}$ are stiffness calculated by AIJ codes and the Ono calculation formula.

Table 9 shows that the calculated results of AIJ codes are relatively safe. Moreover, the difference of bearing capacity along different loading directions under eccentric holes is neglected, thus resulting in the low accuracy. The Ono calculation formula considers the influences of loading direction and reflects the effects of hole position on horizontal bearing capacity along different loading directions. The calculation accuracy of the Ono calculation formula is thus higher than that of AIJ codes with respect to specimens with small holes (SCW-4 and SCW-6) and with large holes (SCW-8 and SCW-9). With respect to stiffness, the bottom slippage of the wall can influence the test results to a certain extent, resulting in the difference in stiffness along the two loading directions. Therefore, the calculation method may differ. In summary, the calculation formulas of AIJ codes and Ono are feasible in determining the influences of holes on the horizontal bearing capacity of SPSW structures. Compared with the calculation results of AIJ codes, those of the Ono formula are closer to the actual test results, indicating its high calculation accuracy. Due to the bottom slippage of walls, the rigidity in the negative direction is lower than the calculated values. In the positive direction, the calculated values are lower than the test results, which is due to the fact that the AIJ code and Ono formula are used for structure design and the result are relatively safer.

## 5. Conclusions

Steel plate-concrete shear walls are studied widely. However, the influence of holes and axial loading on the behavior of steel plate-concrete shear walls are neglected in most studies. Thus, the seismic behavior of steel plate-concrete shear walls is completely different when the influences are considered. Thus, a series of low-cycle reciprocal loading tests were conducted on SPSW specimens with holes in this study. The failure modes and hysteretic curves of SPSW structures under different working conditions were also studied. On the basis of failure modes and hysteretic curves, the seismic behavior of SPSW structures was discussed. Influences of axial compression ratio, hole size, hole position, and the thickness of stiffening plate on the seismic behavior of SPSW structures were considered in the tests. The ultimate bearing capacity of SPSW structures with holes was estimated by AIJ codes and Ono reduction rate calculation methods. The major conclusions are indicated as follows:

(1) SPSW structures without holes mainly develop failures at the roots of shear walls, accompanied with the concrete and bending failures of steel plates. SPSW structures with small holes also develop failures at the roots, but several steel plates crack along the corners of the holes. The failure mode is brittle to a certain extent. SPSW structures with large holes develop significant deformation and better ductility than SPSW structures with small holes but low ultimate loads.

(2) Concrete cracks early under high axial compression ratio. The ultimate strength of SPSW structures increases, but the deformation capacity and ductility decrease to a certain extent. Deformation capacity is enhanced, but the ultimate bearing capacity decreases as hole size increases. With an increase in the thickness of stiffening plates, the ultimate bearing capacity of SPSW specimens increases. Eccentric holes decrease the earthquake-resistant energy-dissipation capacity of SPSW structures, and such a reduction is disadvantageous to seismic design. Therefore, eccentric holes on SPSW structures should be avoided.

(3) The calculated results of AIJ codes are safe, but those of the Ono formula have high accuracy because it considers the influences of loading directions. In summary, both calculation formulas are feasible to calculate the ultimate shear capacity of SPSW structures with holes.

**Author Contributions:** Z.C. and J.W. carried out the experimental research and the academic paper writing. H.L. provided assistance during the experimental research.

**Funding:** This research received no external funding.

**Conflicts of Interest:** There is no conflict of interest.

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
