# Peer review of "Seismic Behavior of Steel Plate-Concrete Shear Walls with Holes"

_applsci, doi:10.3390/app9235255_

Round 1

Reviewer 1 Report

In this paper, a hysteretic curve of specimens was obtained from a low-cycle reciprocal test 12 to discuss the seismic behavior of steel plate–concrete shear wall (SPSW). The effects 13 of axial compression ratio, hole size, the thickness of the stiffening plate, and hole position on the 14 hysteretic performance of SPSW are analyzed. Some critical issues that the authors should carefully check before considering it for publication.
- The abstract should be revised highlighting the novelties.
- The introduction is too long, but the new contributions are not clear.
- The grammatical English should be revised
- Fig 2,3. Should be presented with higher resolution.
- Please mention and comment the prediction of crack and holes in steel plate: https://doi.org/10.1016/j.proeng.2014.06.331,
https://doi.org/10.1016/j.engfracmech.2018.09.032 ,
https://doi.org/10.1016/j.tafmec.2019.102240,

- Table 2 should be organized.
- Line 280: Should add a reference
- Please make sure that the thickness of the plate is 2.5 mm ???
- The authors should be revised the unites of all parameters
- Can the authors predict the defect? (Holes location, size ……)
- In general, the presentation and discussion of results need to be improved.

Reviewer 2 Report

Major points

1.     Figures 2 and 3 are completely unreadable. Authors should provide a more detailed and better labeled figures. In the current size it is impossible to this referee to find where the steel plate and the stiffening plate are located. A thorough description of the specimens is missing.

2.     According to the authors the main objective is to investigate the influence of holes on the wall. However, they have also tested some cases without holes, can they provide comparisons with other researches tests on this type of walls?

3.     Paragraph on lines 294-298 does not make sense. I suggest to the authors to rewrite it.

4.     At Table 9 the differences in the results of the test and of the approximate estimation of rigidity between Negative and Positive values is never explained. Authors should give a reason for these large differences.

Minor points

1.     Captions for Figures 2 and 3 are in red.

2.     Many paragraphs, throughout the paper, have the wrong spacing.

3.     In the test it is mentioned that the ultimate load for the specimen SCW-4 is 1362 but in the Table 4 that value is given for the specimen SCW-5.

4.     In between Figure 5 and Figure 6 there is a figure never mentioned, with no captions and no labeled.

Reviewer 3 Report

This paper presents the test results of steel plate composite shear walls subjected to a loading protocol. Specimens with different configurations concerning thickness, hole area, and compression ratio were considered. The behavior of these plates under loading and their ultimate capacities and displacements are discussed. Finally, the test results are compared to the predictions made by the code and the Ono formula. Please consider the following comments and modify your paper.

- The abstract too short and it doesn't talk about motivation and broader impacts. Please elaborate. Also, is the use of SPSW correct? Please check.

- The motivation for this study provided in the introduction is unclear. There only one sentence on why this study is important in Line 89. Please elaborate on the motivation significantly more, both in the introduction and the conclusions.

- Explain the basis for considering 9 specimens with the listed configurations in Table 1.

- What is the basis for the loading protocol discussed in section 2.3. Please provide references.

- In section 3.3., it is unclear which specimen has how much hole area. Please mention this information in Table 5

- Please clarify whether SCW-1 and SCW-3 overlap in Figure 10.

- I am not convinced if Section 3.4 is thoroughly written. First, why did you provide SCW-1 information in Table 6 when you're not discussing it in the text. Second, the results in Table 6 seem not to agree with the discussion in Lines 234-239. SCW-4 and 6 have somewhat similar strengths in the positive direction. The displacements are not too different as well. This section needs more work.

- In Line 280, provide a reference for the Ono formulae.

- Overall, Section 4 needs some more work in terms of the clarity of expression. You're trying to compare the experimental results with code and Ono formulae and you can simplify how this has been expressed.

- The results presented in Tables 8 and 9 may not suggest that the Ono formula is significantly better than the code one. Please provide average ratios across all the specimens in these tables. Also, amend your discussion between lines 299-329.

- Lines 31-32 seem grammatically incorrect.

- Line 89: Check grammar.

Round 2

Reviewer 2 Report

See attached file
